# Seeking Health in a Digital World: Exploring Immigrant Parents’ Quest for Child Health Information—A Scoping Review

**DOI:** 10.3390/ijerph20196804

**Published:** 2023-09-22

**Authors:** Annina E. Zysset, Patricia Schwärzler, Julia Dratva

**Affiliations:** 1Departement of Health, Institute of Public Health, ZHAW Zürich University of Applied Sciences, 8401 Winterthur, Switzerland; annina.zysset@zhaw.ch (A.E.Z.); patricia.schwaerzler@zhaw.ch (P.S.); 2Medical Faculty, University of Basel, 4056 Basel, Switzerland

**Keywords:** migrants, digital health, information seeking, child health, maternal and child health (MCH)

## Abstract

During pregnancy and early parenthood, parents’ strong interest in pediatric health information presents a valuable opportunity to positively impact long-term health-seeking behavior and overall child health. In line with the increasing prevalence of digital transformation, a scoping review was conducted to explore two key aspects: (1) information seeking and use of digital health information among immigrant parents, and (2) associated reasons and factors. The literature search covered the period until July 2022, using Web of Science, MEDLINE, and CINAHL Complete databases. Out of 625 articles, 12 were included, comprising six qualitative, five quantitative studies, and one review. The majority of studies focused on immigrants in North America, primarily from Latin America and Asia. The studies varied in topics and methodologies, making it challenging to draw general conclusions. Nevertheless, while most immigrant parents rely on digital information on child health, they often prefer human sources such as family, friends, or healthcare providers. Trustworthiness and accessibility emerged as critical criteria for health resources. Two focus group discussions, derived from the results of the review, confirmed these findings for migrant mothers in Switzerland.

## 1. Introduction

During pregnancy and parenthood, mothers and fathers take high interest in pediatric health information, especially in their first parenthood [1]. From a health literacy perspective, this period represents a window of opportunity, as most mothers and fathers are highly motivated to acquire information and competencies given the new responsibility they perceive and carry [2]. Positive experiences with the health system and information received may have a positive impact on future health information-seeking behavior and thus have a long-term impact on children’s (and overall family) health [3,4].

With the increasing digital transformation of health services, modes of health information seeking have undergone a concomitant change. The information sources available to the public have expanded, from traditional (e.g., health professionals, books, print media) to digital modes (e.g., websites, social media). This expansion constitutes a greater magnitude of health information previously unavailable, i.e., prior to digital evolution. Further, while family and friends have always been a source of information, social media has multiplied the number of “friends’” and lay people’s opinions to immeasurable quantities. Recent publications imply that online health information seeking has increased and has become a very popular source [3,5]. An EU survey in 2020 reports that a majority of EU citizens (55%) aged 16–74 had sought online health information related to health promotional or healthcare content in the 3 months preceding the survey [5]. People in Finland showed the highest online health information seeking (77%), followed by the Netherlands (76%), Denmark (72%), and Germany (70%). Studies from Asian countries, including China, the Philippines, Hong Kong, Indonesia, and Vietnam, report even higher rates, above 79% [3]. In contrast, however, despite general high internet usage in Ghana, the awareness and use of digital health information ranged from 15.7% to 31.4% depending on the study [6]. In the USA, based on the Health Information National Trends Survey in 2017, the majority of the population (81.5%) had ever searched for health or medical information, and 70.1% of the participants had reported using the internet first during their most recent search for health information [7].

As information on health is digitalized, the question of digital health literacy and access to digital sources becomes ever-so important. Digital health literacy is defined as “the ability to seek, find, understand, and appraise health information from electronic sources and apply the knowledge gained to addressing or solving a health problem” [8]. Digital health literacy is an evolving concept, which has recently been linked to factors such as motivation [8] and readiness [9]. Access to online information sources can be limited, for example, due to nonavailability of digital devices, connection lag, and unreliability, and unaffordable service costs [6].

Health literacy of immigrants and access to digital sources may be particularly adversely impacted due to migratory processes and status. For example, basic literacy skills, such as reading, writing, and communication, might be very limited in a new language. Furthermore, prior context competencies, such as health system functioning knowledge, or country-specific health norms and practices may no longer be applicable. These challenges likely also affect accessing digital health information, as it relies on similar skills, and thus digital health literacy may equally be reduced. Access to digital health information may also be more limited among immigrants due to their lower economic status in the destination country. However, digitalization may increase transnationalism [10] by sustaining bonds with the home country, and thus reduce the dependency on local sources by permitting access to information sources from the home country or in the mother tongue. In general, the digitalization of health systems may reduce barriers to health information and increase health literacy by more readily providing health information in multiple languages and low-threshold information at no extra cost, as well as making health promotion and information available at any time and place.

### Immigrants’ Barriers to Health Information and Services

Quite a few studies have addressed immigrant-specific access to health information or healthcare. For example, a recent scoping review [11] identified financial and administrative challenges, language hindrances, and unfamiliarity with rights and the system, as well as low health literacy, social exclusion, and direct and indirect discrimination as barriers to healthcare. A Swiss study, however, showed that inequalities regarding doctor visits were more than halved if immigrant groups were similar in socioeconomic characteristics [12].

Some studies specifically investigate barriers of child health access. In Germany, for example, well-child visits (age 0–5 years) and preventive services were less utilized by migrants compared to non-migrants [13]. A scoping review on appropriate healthcare in newborn and young children pointed out that the most relevant factor for accessing appropriate healthcare was cultural sensitivity and understanding of different cultural practices, both as barrier and enabler [14]. Little, however, is known on immigrant digital health information seeking specific to maternal and child health. Thus, we performed a scoping review to investigate maternal and child health-specific digital information-seeking behavior, with the aim to identify reasons and characteristics of use in immigrant populations. Further, the present review served to theoretically inform focus group discussions on the needs and preferences of immigrant women for child health information digitization in Switzerland. For example, currently, the Swiss child health booklet is only available in paper format.

## 2. Materials and Methods

### 2.1. Systematic Literature Research—Scoping Review

For this scoping review, the framework by Arksey and O’Malley was used [15]. It is the most commonly used framework [16] for conducting a scoping review and consists of five key phases: (1) identifying the research question, (2) identifying relevant studies, (3) selection of studies, (4) charting the data, and (5) collating, summarizing, and reporting the results. In addition, we applied the PRISMA guidelines extended for scoping reviews (PRISMA-ScR) [17].

Studies considered eligible for the scoping review were peer-reviewed articles and reviews available in English, published until 27 July 2022; no time restriction in terms of start year of publication was defined. We accepted quantitative, qualitative, and mixed-method studies. A first literature search took place in January 2020 (10 January 2020). A second (24 February 2021) and final, third (28 March 2022) updated search with the same keywords was conducted. The following three electronic databases were systematically searched for articles: Web of Science, CINAHL Complete (via EBSCOhost), and Medline. These databases were chosen after a thematic search for health/health science databases and consulting the university library. The search identified articles referring to the following keywords and corresponding synonyms: “immigrants”, “healthcare”, “digital media”, and “information seeking” in the title and abstract. The final search strategy can be found in the Appendix A. All studies that stated to investigate immigrants were integrated, only studies on refugees were excluded. The results of all databases were exported to Zotero and duplicates removed.

A total of 625 papers were screened for eligibility. Two authors independently screened titles and then the abstracts on relevance and eligibility criteria (see PRISMA flowchart in Figure 1). Papers were considered eligible if they addressed digital information seeking, access to digital media, and digital health literacy in the context of immigration and maternal and child health. Screening results were discussed, and any disagreements were resolved by consensus. The third step included full-text reviews of the remaining 60, nonduplicate articles (see Figure 1). Twelve articles were considered eligible, and data were extracted by two reviewers using an Excel data-charting form. The data-charting form was jointly developed by the project team to organize information extraction. It included bibliographic information on the article and study (author, title, study aim, study design and methods applied, limitations), sample characteristics data (sample size, study population, study country), substantive variables relating to our scoping review questions (examined digital media), as well as resultant findings (general, specific to scoping questions, take home message). The articles were critically appraised for their design and methodological limitations using standardized, appropriate, study-type specific guidelines for critical appraisal [18,19,20]. Two reviewers extracted data independently. The results were continuously discussed and any difference in data extracted was solved by consensus.

### 2.2. Focus Group Discussions

The results of the scoping review informed construction of a focus group discussion guide. The leading question was: How do migrants (mothers) use digital media to inform themselves about (child) health and the health system in Switzerland? Based on the scoping results, general sources of information, reasons for the choice of sources, language of sources and the searches, use of digital sources and tools, and perceived advantages or disadvantages were addressed. The secondary aim was to gain insights on paper or digital preferences of migrant mothers regarding the Swiss child health booklets distributed to every parturient in hospitals or birthing centers.

The discussions were led with available and interested facilitators of regional Femmes-Tische (Women Tables) groups proposed by their coordinator. Femmes-Tische is an educational program for immigrated women and mothers offered in different regions of Switzerland. Based on the principle of “education among peers”, facilitators organize discussion groups with other women from their wider cultural and linguistic background to exchange ideas on topics of disease prevention, health promotion, education, and family. Written informed consent was obtained from all participants. As during these discussions no health-related data were collected from the participants, and as the data were completely anonymized and do not allow for any conclusions to be drawn about the participants, we did not need ethical approval, according to the Swiss human research act.

Two discussions were organized at the end of 2020 with four facilitators from three African and a Latin American country. Facilitators had lived in Switzerland for several years, were in their middle age and had a higher education (nurse, psychologist, veterinarian, engineer), predominantly in the health sector. Three of them had given birth in Switzerland. Each woman represented herself as well as the members of her peer group. The two discussions with an open answer format lasted 53 and 42 min and were analyzed thematically following Braun and Clarke [21].

## 3. Results

### 3.1. Scoping Review Results

#### 3.1.1. Study Characteristics

Of the 12 studies included, six are qualitative, five quantitative, and one is a review. The majority of the studies were conducted in North America, examining immigrants mainly from Latin America (Hispanics, Latinos, Mexicans) and some from Asia (Chinese and Koreans). Three others were from Iran, examining Afghan immigrants; Denmark; and Canada, the latter examining various immigrant nationalities. Equally, the review referred to studies from various countries and nationalities. The parents studied in the articles were heterogeneous regarding their immigration status. While they self-identified as immigrants, a precise definition of immigration status was missing from the articles. Details on the targeted populations are described in Table 1 and Table 2. The topics of the studies were heterogeneous, covering health information-seeking processes, health information sources, and use of digital-health information tools. Details on study characteristics are shown in Table 1 and Table 2. No study was excluded for reasons of limited quality. The main quality limitation related to the cross-sectional study designs, prohibiting causal inferences, as well as nonexperimental, leading to confounding and potential selection biases. Critical appraisals of the selected studies are available as Appendix A.

#### 3.1.2. Information-Seeking Behaviors and Use of Digital Health Information

Most papers assessed the information sources used by their study participants [22,23,24,27,29,31]. Studies that investigated information sources reported that the three main categories of health information sources were (1) healthcare providers, (2) friends and families, and (3) internet/digital sources (see Table 3). However, not all studies provided a ranking of the sources used. All studies addressed the internet specifically as health information source, except for the study by Larson et al. [27]. Very few studies reported digital sources as the main source [24,29]. Results show that immigrant mothers/parents usually use different sources to inform themselves about maternal and child/adolescent health.

Further, more detailed results on information-seeking behaviors and use of digital health information by study are described in the following:

Sharifi et al. [31] investigated information sources and related factors among pregnant Afghan migrant women in Iran. Most of these women were born in Afghanistan (61%), some were born in Iran (39%). Average residency in Iran was 18 years. The most important source was healthcare providers, followed by family and friends, the internet, and media.

In a cross-sectional study, Villadsen et al. [28] examined digital skills, eHealth literacy, and health literacy among pregnant immigrants to Denmark and their descendants (second generation) compared with pregnant women of Danish origin. (Immigrant women were born in 53 different countries, descendants were born in Denmark to mothers originating from 12 different countries and therefore divided into Western- and non-Western-born groups). Western-born immigrants reported lower levels compared to Danish women in use of technology to process health information. Western-born and non-Western-born immigrants reported lower levels in the ability to actively engage with digital services. Non-Western immigrants showed lower levels in the ability to actively engage with healthcare providers and in understanding health concepts and language compared to those of Danish origin. Immigrant women showed lower levels of knowledge and skills in using digital services, which was unrelated to technology access, motivation, or trust. Results were adjusted for age, parity, marital status, and education.

In the study of Mason et al. [24], focus groups were conducted in English unless otherwise requested. No further information on language skills was reported. Immigrant parents in Canada used multiple sources and sometimes different sources to verify the information found. The internet was the primary source to search for child-related dietary or symptom information, followed by friends, family, and healthcare professionals. Further, programs that focus on newcomers and are offered through community organizations, such as Language Instruction for Newcomers to Canada (LINC), were of relevance in order to learn to navigate the Canadian healthcare system, as well as to develop social networks. Parents used the internet to search for children’s symptoms via search engines. They also used internet sites that addressed specific topics such as vaccination and nutrition for their children. The parents had learned to verify information during their own education. They cross-checked the information from different websites, and used websites associated with government or other reputable agencies, as well as books to check the data. Participants were well educated, relatively young and had a high digital literacy [24].

Qian and Mae [25] analyzed forum discussions from a specific chat (“WeChat”) of Chinese mothers in the US on health- and child health-related topics. The participants used digital sources, especially the social media “WeChat”, as a main health information source. Chinese mothers discussed various health topics, such as access to healthcare (e.g., “doctors and hospitals”, “insurance and costs”) and cultural differences between China and the US in healthcare and treatment (e.g., “medicine and treatment”, “alternative healthcare”). The social media platform WeChat enabled Chinese immigrants to facilitate their transnational health information seeking and access to healthcare and to maintain connections over national borders, mainly with relatives and friends in China.

Larson et al. [27] examined health information sources and preferred information materials of immigrant Hispanic women in the US by conducting focus group discussions. Health information sources were provided in Spanish. They relied on human information sources including relatives, friends and neighbors, and healthcare providers. Single individuals mentioned the mass media and the emergency room, but the internet or other digital sources were not mentioned [27]. The majority of the women in this study had a rather low educational level and were unemployed or worked as homemakers.

Criss et al. [23] conducted focus group discussions with Hispanic women in the USA who were either pregnant or had a child below two years of age to assess the information sources used in the first 1000 days (i.e., pregnancy and first two years of a child’s life). The focus groups were conducted primarily in Spanish. Many reported their healthcare provider as trusted information source. Furthermore, all participants used additional sources such as the internet and family members, especially for immediate information. Among family members, the mother was often mentioned as important source about pregnancy and child health, along with other female family members. Some also reported to have received health information from male relatives and few from their boyfriends. Bilingual mothers stated that they got much of their information from their own parents. The internet (e.g., Google, BabyCenter.com) was reported in all focus groups as health information source. They searched for topics such as healthy fetus, child development, and nutrition and safety information. Mainly English sites were used. The site BabyCenter.com was well known among participants: many signed up for weekly emails and downloaded the corresponding app. In the group of pregnant women, weekly updates on development of child and health information during pregnancy were preferred. On YouTube and Facebook, videos containing child health information (e.g., nutrition during pregnancy, baby development, breastfeeding) were watched. However, the majority did not trust the information on these channels. Desire for accurate information was stated in all groups. Often, doctors/pediatricians were asked for advice to verify digital information.

Lee [29] compared the health information-seeking behavior of US and recently immigrated Korean mothers living in the USA. Both groups used the internet frequently to search for information about child health but also used a variety of other sources like healthcare providers, nurses, friends with children, husbands, or other relatives. Overall, the frequency of information searches was higher in Korean mothers and the preferences of sources were slightly different between the two groups. American mothers showed a tendency to prefer human sources, while Korean mothers equally liked human and nonhuman sources. In terms of digital sources, blogs and online forums were among the most used sources of Korean mothers, while they ranked lower among American mothers.

Gonzalez et al. [26] examined the interesting setting of intergenerational online health information seeking of Latino families in the US, where bilingual, teen-aged children (10 to 19 years) helped their parents to search online for health information. Often, the technical and/or English language skills of parents were limited, and children could support them in finding and interpreting the information. Parent–child dyads applied collaborative search strategies, using smartphones, tablets, laptops, desktop computers, and combinations of different devices to search for information. They used different internet browsers, search platforms, and mobile applications (e.g., Google Maps (Google, Mountain View, CA, USA), Google Translate (Google, Mountain View, CA, USA)).

Silverman-Lloyd et al. [30] examined the use of interactive text messages throughout a child’s first year in a low-income, limited English proficiency Latino population in the USA. The acceptability of the text messages was very high and rated as easy. Parents reported that the messages helped them remembering their appointments at the clinic (97.2%), made them feel more connected to the clinic (95.8%), and gave them a feeling of support from the clinic (94.4%). Participants with higher education engaged more with the text messaging intervention than participants with lower education.

Reuland et al. [32] examined access to information and communication technology and use of common applications by low-income immigrant Latino parents, who preferred healthcare language in Spanish, in the USA. The Spanish-language survey was completed through guided oral administration by a bilingual (English/Spanish) researcher. Nearly all participants (91.7%) had a smartphone, but only few had a phone plan with included unlimited data access (24.2%). Only 39.5% had access to internet at home. In terms of access to technology, 22.3% were categorized as having low access, 47.1% as having moderate access, and 30.6% as having high access. Family income was associated with the access category. In terms of use, 82.2% reported frequent use of text messaging. However, frequent use of health-related applications was lower than for all other applications (8.9%).

Recto and Champion [22] assessed information seeking for perinatal depression of Mexican American adolescent mothers. All adolescents were born in the United States and were highly acculturated. The majority spoke English at home (55%) and with friends (70%), while a small proportion indicated speaking both English and Spanish equally (25%). Information sources mentioned were internet search engines (e.g., Google), social media, websites on depression, libraries, healthcare providers, and mothers’ mothers who experienced depression themselves. Frequency or the most used sources were not assessed, but the family support systems (their mothers in the first place) seemed to be most important for support for the adolescents. When family support lacked, friends and the father of the baby acted as the support system. Interestingly, most adolescents stated not to know how to verify information from the internet. Individuals suggested reading reviews by others who visited the website or verifying the information with a healthcare provider [22]. It is noteworthy that adolescents were recruited from parenting classes and had already been provided with valuable information; therefore, the sample might not be representative or might be biased.

Hughson et al. [33] conducted a review on pregnancy apps and how these are used by culturally and linguistically diverse women, as well as by women in general (i.e., women living in their birth country and speaking the country’s main language). They analyzed the utility and quality of information provided. One result was the lack of literature on pregnancy app usage among culturally and linguistically diverse women. Only one study on this topic was found [34]. In general, pregnancy apps are very often used by pregnant women.

#### 3.1.3. Reasons and Factors Associated with Digital Health Information Seeking

Higher or more frequent digital health information seeking was suggested by study authors to be associated with factors such as higher education [26,30,31], high digital literacy [26], health literacy [28], not having health insurance [24,31], having the first child, respective lower number of children [29,31], access to the internet/having a personal device to access the internet [32], and less years lived in the country [29] vs. more years lived in the country [31].

Information sources that are easily accessible and convenient were preferred [24], depending on the circumstances and skills of the searchers and on the preference of digital or personal information sources. Avoiding waiting a long time for a doctor’s appointment was reported as a reason for using the internet as information source [23,24]. Additionally, parents not speaking English (i.e., main language of country) have problems communicating with the doctors [24]. In this case, family members often help to translate [24].

Lower education was associated with less likely use of healthcare providers’ health information in the study by Lee [29]. In the study by Sharifi et al. [31], women had a low educational background and chose healthcare providers most often as information source (65.1% vs. family and friends 47.5%). Higher education was associated with higher technology access among low-income Latino parents [32]. In the study by Mason et al. [24], participants were well educated, relatively young, and showed good subjective digital literacy, which might explain the primary use of the internet.

The reasons for information-seeking choices were not assessed and not explicitly stated in the study by Recto and Champion [22]. Apparently, participants searched for information when they were personally affected by a health problem (e.g., perinatal depression) and needed information or assistance [22]. In the study by Criss et al. [23], women used digital sources to find information about various pregnancy and child health topics. Quicker availability was stated as a reason for preferring the internet over healthcare providers. 

### 3.2. Focus Group Results

All four participants of the focus group discussions stressed that in our day, women tend to seek information on the internet via a Google search. In some countries (e.g., Egypt), mothers with newborn or young children organize themselves in Facebook groups, moderated by competent health personnel, where they can post questions and receive trustworthy answers. Emigrated Egyptian women may continue to participate in these groups. Many women also rely on family support systems and call their own mothers, sisters, other family members, or friends in their home countries for advice, family medicines, and positive experiences. Less frequently, husbands are asked for advice. Others resort to their healthcare provider (pediatrician, nurse, pharmacist) of maternal counseling centers or children’s hospitals. Elderly women also consult books on pediatric development or diseases.

In Switzerland, child health information is distributed via a child health booklet every parturient receives each time she gives birth in a hospital or birthing center [35]. This booklet is available throughout the country in the three main national languages (German, French, Italian) and in English; it is provided by the Swiss pediatric society and allows for personal notes on a child. The participants consider it a very valuable base of information and use it as an index of terms, sometimes translated in their languages, for additional Google searches. The language of the health booklet was not an issue among the participating women. When asked if they prefer this booklet in printed or in digital format, the spontaneous answer in both discussion groups was the preference for the physically available printed version: “We are paper women, aren’t we?”, one middle-aged woman commented. The cited advantages were especially the possibility to hand it over to the grown-up child as a souvenir along with a photo album, the possibility to make their own notes, the safety in the sense that it was perceived as being lost less easily compared to digital data. In the further discussion, however, the advantages of a digital version were cited, like the availability at any time and any place and the fact that it could be shared electronically with all persons involved in the care or having responsibility for a child. The women stressed that since the COVID-19 epidemic, people have gotten used to digital information sharing (“we are always digital”, “without digital media, we cannot advance”). At the end, in both discussion groups the consensus was the preference for the availability of the booklet in printed and digital formats to maximally benefit from the advantages of both versions.

## 4. Discussion

This scoping review aimed to examine digital information-seeking practices regarding child health topics by immigrant parents. We found only few studies investigating the combination of maternal and child health, digital information seeking, and immigrants, indicating the novelty or disregard of the topic in this specific age group. The oldest study was published in 2009 whereas most studies were published in the last five years (since 2018), underscoring the novelty of the topic. Given the heterogeneity of the studies included, it is rather difficult to deduce general conclusions. Most studies, however, showed that while immigrant parents use digital information to inform themselves about child health, human sources—family/friends or healthcare providers—are often preferred. Overall, trustworthiness and accessibility are important.

### 4.1. Digital Sources

Our scoping review shows that immigrant mothers and parents frequently use digital information sources. Most often, internet searches were conducted via search browsers or specific websites were consulted. This corresponds to the general search behavior on health topics found for the general population and for parents. A review on health information seeking, not specific to immigrants and child health, showed that 83% searched via general search engines and 15% via specific health information websites [3]. Studies on information seeking of nonimmigrant parents also mention frequent use of the internet for child health information [36,37]. Other digital sources reported in our review were online forums, blogs, apps, or specific chats. Social media were primarily used to connect with other mothers more than to search for health information. However, this latter use of social media does exist, as described in the studies by Recto and Champion, Criss et al., and Lee [22,23,29].

Pregnancy apps are often used to access fetal development information [33]. However, Hughson et al. found that data on the use of these apps by immigrant women are scarce [33]. Some findings regarding digital sources might be specific for certain nationalities and the immigration country. WeChat (Chinese instant messaging, social media, and mobile payment app developed by Tencent, Shenzhen, Guangdong, China), for example, is a Chinese chat forum used by Chinese immigrants as well as Chinese persons living in China [25]. Given that most studies were from the USA, the frequent use of digital health information may risk being overestimated. Not every USA citizen or immigrant has health insurance, and digital information sources might be used more frequently to avoid a health provider visit and costs. On the other hand, high use of digital information sources by parents was also found in studies conducted outside the USA [24,36,37].

Studies repeatedly indicate a preference for human over digital sources. While in some studies healthcare providers were the first/most used source [23,31], other study populations relied more on family/friends [22,27]. The personal sources of information often include a mother’s own mother or female friends/relatives [22], sometimes also personal contacts in the country of origin, including family members [24,25]. Gonzalez et al. [26] report that migrant parents also rely on their digitally literate children, more often on daughters than sons, to help them with digital information searches, e.g., to interpret acute symptoms of younger siblings [26]. Reasons for the preference of human sources are often trustworthiness and accessibility, as stated above. Moreover, the study by Recto and Champion [22] showed that most female adolescents did not know how to verify information in the internet. This insecurity for verifying information found in the internet was frequently reported in this review [23,28] and other studies [38] and is often stated as a reason to consult health professionals to verify the information found [23,38].

### 4.2. Access to Digital Information Sources

As in many industrialized countries, digitalization is advanced and a high percentage of the populations have access to and use the internet (Europe 87%, USA 81%) [39,40]. All but one study, Larson et al. [27], reported the use of the internet to access information. Although digital devices and the internet are considered widely used, authors describe a digital health divide. Reuland et al. [32] concluded from their study of Latino parents in the USA that they are often smartphone-only users with limited data access plans. Thus, mHealth interventions (utilization of mobile devices for delivering healthcare services and information) with need of high data volume, sent via email or offered via apps, may not reach all target individuals, especially not persons with low income. This finding might also apply for other low-income parents. In Denmark, the gap in digital health access for immigrant women is attributed to their reduced ability to interact with healthcare providers and navigate digital services, posing an additional challenge in obtaining digital health information [28].

Language difficulties are a barrier to understanding of health information or a health system [11]. It has been hypothesized that digitalization of the health information and system can reduce this barrier. Indeed, some studies indicate that migrants prefer information sources in their own language, and that this information is accessible and available [11]. Partly, immigrants resort to digital communities in their country of origin, which may sustain transnationalism [10] and slow down integration into the new health system. An example is WeChat, the most popular social media platform in China and among Chinese persons around the world. It provides multiple ways of sharing information and of networking, including posting information with photos and communicating privately or in groups [25]. In this chat, recommendations for local Chinese doctors are often shared because of language facility and a preference for physicians of their own culture [25]. Another example is the Facebook groups organized in Egypt and moderated by health personnel cited by a participant of our focus group discussion. For large immigrant groups, health information is partially translated, as the language issue is well known. For example, there is plenty of Spanish health information in the USA for the Hispanic population [27]. As Larson et al. have shown, not only the language but also the design of the information must be adapted accordingly [27].

Apart from access to digital devices, digital literacy or digital health literacy emerged as a theme. Varying levels of both digital literacy and health literacy can present a barrier to searching online for health information or using digital tools to access healthcare [3,4,28,36,41]. Villadsen et al. [28] observed that pregnant immigrant women in Denmark showed lower levels of digital skills and eHealth literacy compared to women of Danish origin, which may have led to the lower use of digital information. The study by Gonzalez et al. [26] showed that such barriers were overcome with the help of digitally skilled offspring. The reliance on children’s skills in the context of family health (e.g., acute disease of younger siblings), to find and to interpret relevant health information puts a high responsibility on the children [42]. Moreover, younger and digitally native people may be skilled in the use of digital devices; however, this does not automatically correspond to a high level of digital health literacy [43].

More sustainable than relying on family or friends would be to educate migrant parents and increase their own digital health literacy. As Denmark is digitally advanced, increasing digital skills of immigrant women was proposed by means of specific courses. Villadsen et al. [28] found small positive effects in a study sample consisting of many low- and middle-income women. Lower digital health literacy has been associated with lower education [44]; thus, the results in this study might be partly due to lower education rather than uniquely due to the immigrant status.

Lower education, language difficulties and lack of knowledge of a healthcare system are likely to cumulatively interact, resulting in reduced access to health information and services and thus in worse health outcomes [45,46]. Results from studies on immigrants and digital health information seeking, not limited to child health, showed quite consistent results for the association between higher education, higher socioeconomic status, younger age, and frequent use of digital sources [47,48,49,50]. These factors apply for native populations, too [3,36,37,51]. Regarding education and socioeconomic status, studies in this scoping review are consistent, with one exception. Rather counterintuitively, lower education was associated with less likely use of healthcare providers health information among Korean mothers in the USA [29]. While results on the association of education and use of digital sources were frequent, age has not directly been examined as a factor. For example, in the study by Mason et al. [24], participants were relatively young, well educated, and showed good subjective digital literacy, which resulted in frequent internet use.

Next to socioeconomic status, other individual characteristics like number of children were associated with digital information seeking. The two studies addressing the number of children [29,31] yielded that primipara search more frequently for digital child health information than multipara. Mothers who have already given birth once may be less likely to search for new information and more likely to refer to known sources used for the first child. Pregnancy and infancy might therefore be a good time period to educate first-time mothers/parents in digital health literacy, and thus enable them for safe and helpful future digital information seeking behavior.

The findings of the scoping review are largely supported by the focus group discussions held in Switzerland. One exception relates to language being a hindrance to local health information. The group did not see language as a barrier, which might be due to the higher educational level of the participants. Interestingly, the women mentioned a more recent change in health information-seeking behavior. For one, they indicated a potential generational change in digital health information seeking, referring to themselves (middle aged) as “paper women”, while younger women were considered or perceived themselves more digitally skilled. Secondly, they pointed to an increased acceptance of the digitalization of health system and information due to the experiences during the COVID-19 pandemic. For this reason, both a paper and a digital version of the Swiss pediatric health booklet were qualified as useful.

### 4.3. Limitations and Strengths

While different search terms for the status “migrant” were used, it is possible that we missed studies addressing migrants’ digital health information seeking in population-based studies. Overall, we did get the impression that the specific topic of child health was underrepresented. Most studies included are relatively new, the oldest having been published in 2009, indicating that the topic is being addressed more in recent years corresponding to the digitalization of societies and healthcare. The results are only partly generalizable due to the heterogeneity of migrant nationalities and origin of the studies. Most studies were conducted in North America, only one each in Europe and Asia. The scarcity of studies in Europe and Asia might relate to a later start of the digital transformation. The number of participants in the focus groups, taking place during the COVID-19 pandemic, was small. However, the women were facilitators of “Femme Tische” groups, thus referred not only to their own experiences but also to the experience made with a number of immigrant women living in Switzerland.

## 5. Conclusions

This scoping review indicates a generally high awareness of digital health information sources among migrant parents, albeit demonstrating a large variety in its actual use and application of health information. The preference for and trust in health professionals, as well as in family members’ advice regarding health issues, remains high. Given the higher affinity and digital literacy in younger women and mothers, it is safe to assume that the use and need for digital health information will increase and practice should address this generational change. Factors such as educational level and availability of digital devices prove to be associated with higher levels of digital health information seeking. Furthermore, increased use of digital health information and digital health literacy is positively associated with a higher stage of sociocultural integration. Given these factors, there is a risk of further disadvantage of certain immigrant women by increasing digitalization of health services and information. Providing health information in simple language, irrespective of immigrants’ educational level, improving accessibility via common digital devices and low data volumes, and providing culturally adapted content seem relevant to ensure equitable access. The scarcity of publications indicates a lack of research, especially in Europe. Given the chances and risks related to the use of digital health information sources in the new country as well as the country of origin, more research seems warranted. The life stages of pregnancy and maternity could be a window of opportunity to reach immigrant women with maternal and child health information and establish higher digital health literacy.

Accordingly, and according to the consensus of the focus group discussions with facilitators of regional Femmes-Tische groups, the child health booklets distributed to every parturient in hospitals or birthing centers in Switzerland addresses this window of opportunity and should remain available in print but also be offered in a digital format to allow migrant mothers to maximally benefit from the advantages of both versions.

## Figures and Tables

**Figure 1 ijerph-20-06804-f001:**
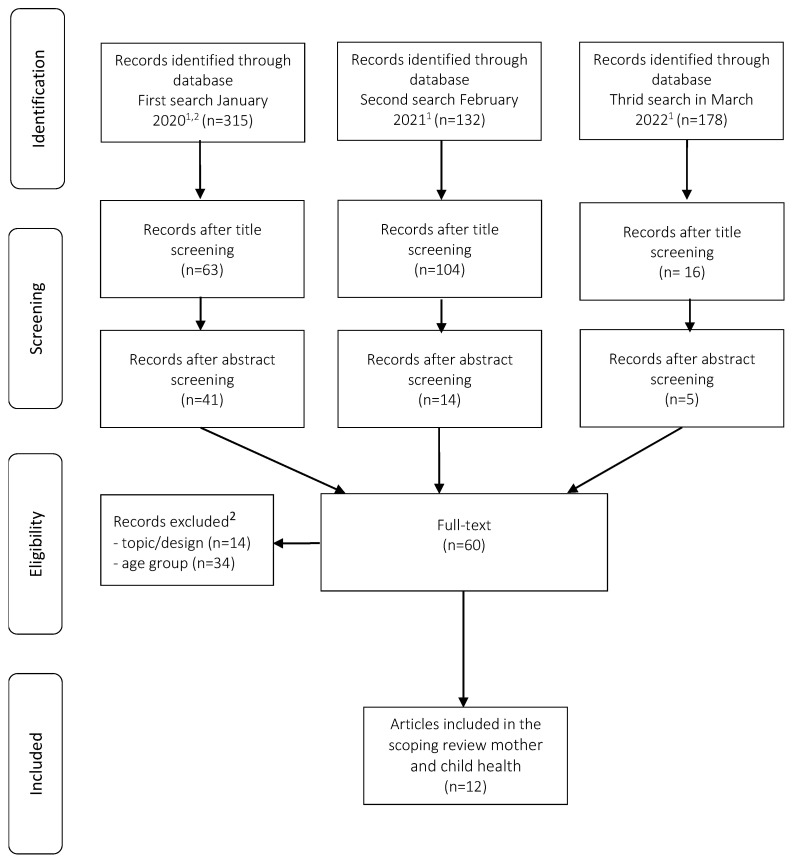
PRISMA flowchart. Notes: ^1^ Results after removed duplicates. ^2^ The initial search 2020 contained articles on age groups CAH (child and adolescent health) and ADH (adult health).

**Table 1 ijerph-20-06804-t001:** Characteristics of included articles by study type—qualitative studies.

Author	Country	Aim	Qualitative Method	Sample Size	Targeted Population (IC = Inclusion Criteria)	Migrants	Information Seeking
Recto and Champion 2018 [22]	USA	Examine mental health literacy of Mexican American adolescents, i.e., perception of cause of perinatal depression, knowledge of self-help strategies, how they obtain information.	Interviews	N = 20	Mexican American female adolescents IC: pregnant or postpartum ≤ 1 year; between 15 and 19 years of age, self-identified as Mexican American	Primary aim	Secondary aim
Criss et al., 2015 [23]	USA	Explore how health information sources inform decision-making among Hispanic mothers during their first 1000 day of life, and to generate appropriate health information sources and communication strategies for future interventions.	Focus groups sorted by life stage: pregnancy, infancy (0–6.9 month), early childhood (7–24 months)	N = 49	Hispanic women IC: self-identified Hispanics, able to speak Spanish or English, >18 years, having a singleton pregnancy or at least one child <2 years without a major medical condition.	Primary aim	Primary aim
Mason et al., 2020 [24]	Canada	Understanding immigrant parents’ health information-seeking across ethnically diverse groups of immigrants.	Interviews	N = 50	Immigrant families in western Canada	Primary aim	Primary aim
Qian and Mao, 2021 [25]	USA	Explores how Chinese immigrant mothers use the ethnic social media—WeChat—to engage in health information sharing and coping with cultural differences in healthcare between the US and China.	Analysis of chat discussion	N = 253 mothers/N = 89 discussions	Chinese immigrant mothers in WeChat group.	Primary aim	Secondary aim
Gonzalez et al., 2020 [26]	USA	Understanding of how family members work together to access online health information (intergenerational technology and health information brokerage)	Interviews with parent-child dyads and observation of online search tasks	N = 24 parent-child dyads	Latino immigrants IC: the parents self-identified as Hispanic/Latino and were born in Latin America, had a child between the ages of 10 to 17, reported that their child helped them search for and/or translate online information.	Primary aim	Primary aim
Larson et al., 2009 [27]	USA	How are health-related messages obtained? Which educational materials available in Spanish are preferred?	Focus groups	N = 26	Hispanics IC: female head of household, Spanish as first language, born outside the US, having participated in the clinical trial for at least 11 months (to assure exposure to various educational materials).	Primary aim	Primary aim

**Table 2 ijerph-20-06804-t002:** Characteristics of included articles by study type—quantitative studies and review.

Author	Country	Aim	Study Type	Sample Size	Targeted Population (IC = Inclusion Criteria)	Migrants	Information Seeking
Villadsen et al., 2020 [28]	Denmark	Exploring eHealth literacy, ability to actively engage with healthcare providers and health system navigation among pregnant immigrant women.	Quantitative	N = 405	Pregnant immigrant women and their descendants compared with women of Danish origin. IC: being pregnant, having the ability to reply to the questionnaire in Danish or English.	Primary aim	Secondary aim
Lee, 2018 [29]	USA	Examine health-related information need, seeking, and sources preferences between two distinct groups of mothers.	Quantitative	N = 480	US and immigrant Korean mothers of healthy children in the USA IC: 18 years or older, mothers with at least one child under age of 3 years, born in US with US citizenship or born in Korea and moved to US within past 10 years, child’s doctor visits in past 3 months lower than 3, child never been treated by a specialist.	Primary aim	Primary aim
Silverman-Lloyd et al., 2020 [30]	USA	Evaluating the feasibility and acceptability of interactive Spanish language text messages sent throughout a child’s first year of life.	Quantitative	N = 79	Latino immigrant parents IC: parents/legal guardians, singleton infant <2 months of age, enrollment in public health insurance, parent age >18, self-identification of Latino/a, foreign-born, parent preferred healthcare language of Spanish, and at least one household cellular phone.	Primary aim	Secondary aim
Sharifi et al., 2021 [31]	Iran	Investigating the sources of information and its related factors among pregnant migrant women.	Quantitative	N = 280	Pregnant Afghan migrants IC: current pregnancy, Afghan nationality, and the presence of a health record at one of the four involved health clinics.	Primary aim	Secondary aim
Reuland et al., 2022 [32]	USA	Examine access to technology and use of information and communication applications/programs by immigrant Latino parents of infants to guide mHealth developments.	Quantitative	N = 157	Latino Immigrant Parents IC: parents/legal guardians of publicly insured, singleton US-born infants, <2 months of age, minimum respondent age of 18 years, self-identification of Latino/a, foreign-born, preferred healthcare language of Spanish, and at least one working cellular phone in the household.	Primary aim	Secondary aim
Hughson et al., 2018 [33]	not applicable	Review of the literature on pregnancy apps and aims at describing (1) the ways in which apps are used by women, in general, and by those of a CALD background; (2) the utility and quality of information provided; and (3) areas where more research, development, and oversight are needed.	Review	N = 38 Studies	General female population and culturally and linguistically diverse (CALD) female population.	Secondary aim	Secondary aim

**Table 3 ijerph-20-06804-t003:** Results by study country.

Country, Continent	Author	Sample, Origin, Age	Digital Media Examined	Health Topic	Summary of Main Findings	Results Related to (im-) Migrants and Digital Information Seeking	Nationality
Iran, Asia	Sharifi et al., 2021 [31]	Pregnant Afghan migrants, mean age = 29.6 years.	Internet (e.g., websites, Facebook) vs. media (e.g., radio, TV, newspapers) and healthcare providers (e.g., midwives) vs. family, friends, and relatives	Health literacy/maternal health/pregnancy	Most important sources of information: healthcare providers (65.1%), family and friends (47.55%), the internet (32.1%), and media (18.9%). Significant factors: education level, number of children, length of residence in Iran, place of birth, and insurance status.	Factors associated with higher internet use: lower number of children (* < 0.001), higher education level (* < 0.001), birthplace Iran (* 0.001), longer time residing in Iran (* < 0.001), and no insurance (* 0.008).	Afghan
Denmark, Europe	Villadsen et al., 2020 [28]	Pregnant women, immigrants, and their descendants compared with women of Danish origin. No age limitation, age category of majority of participants per migration status is Danish origin: 55.7% = ≥31 years; non-Western descendant: 64.7% = 25–30 years; non-Western-born immigrant: 47.3% = ≥31 years; Western-born immigrant: 69.4% = ≥31 years.	Nonspecific	Health literacy	Overall trend for lower ehealth literacy and health literacy among immigrants and their descendants compared with women of Danish origin. Regarding ehealth literacy, challenges related more to digital abilities than motivation, trust, and access to technology.	The mean ability to engage with digital services was 3.20 (SD 0.44) for women of Danish origin. Non-Western descendants (−0.14, 95% CI −0.31 to 0.02), non-Western (−0.20, 95% CI −0.34 to −0.06), and Western (−0.22, 95% CI −0.39 to −0.06) immigrants had lower adjusted means of this outcome. No differences In motivation to engage with digital services were found for descendants (−0.00, 95% CI −0.17 to 0.17), non-Western (0.03, 95% CI −0.11 to 0.18) or Western (−0.06, 95% CI−0.23 to 0.10) immigrants compared with the mean of the reference (2.85, SD 0.45). Lower ability to engage with healthcare providers was found for non-Western-born immigrants (−0.15, CI 95% −0.30 to −0.01) compared with the mean of women with Danish origin (4.15, SD 0.47).	Various
Canada, North America	Mason et al., 2020 [24]	Immigrant families in Western Canada, mean age = 38 years.	Nonspecific	Child health	Three main themes: Accessing social networks for informational support, role of professionals in accessing healthcare information, navigating, and evaluating information sources. Immigrant families consult various sources of information to meet their children’s healthcare needs.	For immigrant parents, the internet was the most preferred means of accessing information (e.g., checking symptoms). The second most sought sources of information were participants’ peers and relatives (e.g., friends and family), and the third doctors (family doctors, pediatricians). Participants mainly used sources of information easily accessible, convenient, and in close physical proximity first.	Various
USA, North America	Qian and Mao, 2021 [25]	Chinese immigrant mothers in WeChat group	Chat forum	No specific/health information sharing	Frequently discussed topics were “doctors and hospitals”, “insurance and cost”, “medicine and treatment”, and “alternative healthcare”. Participants constantly compared Chinese healthcare beliefs and practices with western ones. They adopted various acculturation strategies to manage the cultural differences in healthcare beliefs, practices, and systems.	Immigrant Chinese mothers used the Chat forum to get various health information, such as access to health service, costs, suggestions for doctors. Over one third of discussions (N = 38) were about children’s health, followed by general family health (N = 18), few on grandparents (N = 5). Many discussions (*n* = 22) compared Western and Eastern medical practices to further understand both systems. There was a preference for Chinese medical beliefs (*n* = 12) compared to American ones (*n* = 2). More than half (*n* = 52) were on information/suggestions on doctors and patents. They preferred to communicate in Chinese in the Chat.	Chinese
USA, North America	Larson et al., 2009 [27]	Hispanics, age range = 22–45 years.	Nonspecific	General health information	Primary sources of health information relatives and friends (42.9%, 6/14), clinicians (35.7%, 5/14), media (14.2%, 2/14), or emergency room (7.1%, 1/14). Two factors associated with effective information materials: purpose-based information assessment and design of materials.	Preferred educational material used either a question and answer or true/false format and could be shared with and used to teach children. These ones were preferred over the ones with colorful design and pictures or lower reading levels. Internet was not cited as source of health information and infrequent use of computers was reported.	Hispanic
USA, North America	Criss et al., 2015 [23]	Hispanic women, mean age = 26.4 years.	Internet, mobile apps, social media (Facebook, YouTube)	First 1000 days	Trusted health information sources included healthcare providers, female and male family members, internet sources (e.g., BabyCenter.com), selected social media, and TV shows. Women expressed desire for accurate information, inclusion of extended family in communication and intervention. For validation of information, they compared different sources; thereby, the doctor’s advice was important.	At all three life stages (pregnancy, infancy, early childhood), women used the internet (Google and specific websites) as health source for child development, nutrition, feeding practice, and safety information. Some used social media (YouTube and Facebook), but majority did not trust information on Facebook. Interest in receiving reliable website links from doctors was stated. For further interventions, the authors suggest the following health information sources, provided by healthcare providers or public health professionals: Email with tips and links, text message via smartphone and apps, telephone hotline, mail/brochures with health education material, and classes.	Hispanic
USA, North America	Lee, 2018 [29]	US mothers and immigrant Korean mothers in the USA of healthy children, age range = 31–35 years.	Internet, social media, blogs compared to personal and professional information sources	Child health	Ninety-three percent of the US mothers and 94.2% of immigrant Korean mothers had searched for health information in the past six months. US mothers preferred human sources (e.g., doctors, nurses, their husband, and other relatives), whereas immigrant Korean mothers preferred nonhuman sources (e.g., online communities, books). Both groups searched most frequently for information about diseases.	Most frequently used source in both groups was the internet (US mean 3.81, Korean mean 4.49; scale 1–5), followed by blogs and online forums for Korean mothers (mean 4.03) and friends with kids for those US born (mean 2.92). Koreans used family (M = 2.94) more frequently than healthcare providers (M = 2.79). Mean usage frequency of sources overall was higher among Koreans. Higher social media use was associated with older age (OR = 0.03 and 0.23 for 18–30y and 31–35y, reference = 36 + y), less years lived in the US (OR = 0.75), and number of children 1 vs. >2 (OR = 2.11). Having one child was associated with overall more frequent use of most sources.	Korean
USA, North America	Gonzalez et al., 2020 [26]	Latino immigrants, median age = 41 years.	Internet/use of technology, different devices for access	Health information/child health	Intergenerational online searching and brokering reveals an opportunity to promote health literacy in culturally relevant ways. Female preponderance in the survey generates a hypothesis on gendered behavior.	Technological skills transfer and intergenerational brokerage in bilingual families increase access to health information and family eHealth literacy. Findings indicate that Latino immigrant families regularly use ICTs to collaboratively access online health information. Health literacy skills and strategies are engaged and shared by parents and children as they collaborate to find, comprehend, and use health information to make informed choices.	Latino
USA, North America	Silverman-Lloyd et al., 2020 [30]	Latino immigrant parents, mean age = 30.1 years.	Smartphone/text message	Child health	Majority of participants (93.1%) rated the usability of the text messages as very easy/easy. Messages helped them remembering their appointments at the clinic (97.2%), made them feel more connected to the clinic (95.8%), gave them a feeling of support from the clinic (94.4%), and helped them with obtaining medications (65.3%).	Participants with higher education engaged more with the text messaging intervention than lower educated ones; 73.7% in high engagement category had completed at least high school compared with 20.0% in the low engagement category, *p* = 0.002.	Latino
USA, North America	Reuland et al., 2022 [32]	Latino immigrant parents, mean age = 29.4 years.	Technical devices and internet	No specific	mHealth interventions that use data, email, or an application interface may not have the intended reach or effectiveness among low-income immigrant Latino parents, but interventions that employ text messaging may be more effective. Smartphone internet access only was frequent and limits the reach and potential of internet-based interventions. Limited literacy, including general, health, and digital literacy, is likely to impact technology use and acceptance.	Total 91.7% of parents owned a smartphone and 60% accessed the internet only via their smartphone. Around one-quarter (24.2%) of participants had access to unlimited data. Frequent use of text messaging was common (82.2%), but frequent email use (about 40%) was less common. Less than 10% of participants frequently used health-oriented applications. Health apps were the least used, with no association to technology access category (low to high tech access). High tech access parents used more frequency email than other groups. Access to internet and digital tools must be considered when designing health interventions for low-income immigrants.	Latino
USA, North America	Recto and Champion 2018 [22]	Mexican American female adolescents, mean age = 17 years.	None	Maternal depression beliefs/sources	Interpersonal conflicts with family or partner and transition to motherhood were seen as causes of perinatal depression. Knowledge and beliefs of self-strategies included activities that distract from depression, like going out with friends and turning to others for support like own mother, father of the baby, school, parenting classes.	Mentioned information on perinatal depression: search engines (e.g., Google), social media, depression websites, libraries, healthcare providers, and mothers with experience with depression. Adolescents were not sure how to verify information from the internet. No preferred source was specified among the above mentioned, but mothers seem to be especially important.	Mexican
Not applicable	Hughson et al., 2018 [33]	General female population and culturally and linguistically diverse (CALD) female population	Pregnancy apps and other digital devices (mobile phone, internet, general apps)	Maternal/pregnancy health	Pregnancy apps were often used to access pregnancy health and fetal health in general population. Further popular app features were data storage capability, web-based features, or personalized tools, and social media features were also popular. Women reported desire for pregnancy apps that are relevant to their local healthcare context and come from a trusted source. Quality standards in app industry are missing, but reliable sources are desired by users.	Studies on the use of pregnancy apps in CALD population context are missing. Some evidence exists that CALD women use digital sources for pregnancy information. Lower-income and technological, health literacy, and language issues might be indicators of lower use of pregnancy app among CALD women.	Not specified

## Data Availability

Not applicable.

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
