# Peer review of "Seeking Health in a Digital World: Exploring Immigrant Parents’ Quest for Child Health Information—A Scoping Review"

_ijerph, 2023, doi:10.3390/ijerph20196804_

Round 1
Reviewer 1 Report
Thank you for the opportunity to review this paper. The review has been well designed, but it is not clear why this was limited to three databases. There are other databases with a large number of health papers that may include articles relevant tot his topic. The stated databases used differed between the abstract and the methods. Was ethics approval sought for the focus group and was consent captured? What do you mean by “were analyzed content wise”? Were they thematically analysed? The results for the review are clearly presented, though it would benefit from sub-headings to guide the reader. The focus group results is very limited and it is unclear how the results of the review guided the focus group discussion – the two seem very separate. What themes emerged from the focus group? How will this inform practice? The Discussion is well crafted and the only area for improvement would be greater discussion about what this means for policy and practice.
Reviewer 2 Report
Very interesting and important topic. Paper could be improved by a more exact definition of immigrant (first generation only? including immigrants in early childhood as well?) - many studies define immigrants up to third generation but distinguish - would be important to note own definition as well as definition used in articles. For example, Criss article in review doesn't seem to define population by immigration generation at all.
More accurately, article should state immigrant populations primarily from Asia and Latin America (including Mexico, Central America and South America), shouldn't use South America as umbrella term. Description of Danish immigrants as coming from Western or not countries confusing (be more exact).
Wonder if there is a way to quantify educational levels more - or compare general educational levels by country when discussing digital literacy among immigrants? In my experience with adolescents, immigrants from Korea, China, VietNam have much higher literacy levels (even if pre high school level) than immigrants from rural regions of Latin America) and this translates to more digital fluency (e.g. ability to use translation apps) if their English is limited

Generally good. There is an awkward use of "mention" on page 29.
Round 2
Reviewer 1 Report
Thank your for your revisions
Author Response
Thank you very much, we are glad that the revision was to your satisfaction.
Reviewer 2 Report
Appreciate the author's corrections. My only remaining concern is that the concept of immigrant generation is not well-defined. I was most concerned in original review, for example, about inclusion of review by Cress et al. that did not define immigrant generation (self-defined Hispanic families may have been living in the US for hundreds of years) - I actually read the Cress article on second review, and although the authors don't define immigrant generation, the focus groups were conducted primarily in Spanish, and bilingual mothers stated that they got much of their information from their own parents. As a reader, I would appreciate some indication of this, e.g. bilingual, or focus groups in Spanish.
Improved, there are still minor errors, e.g. use of prepositions, that don't affect the meaning of the article but are still jarring to native speakers of English.
